# Efficacy of Integrase Strand Transfer Inhibitors and the Capsid Inhibitor Lenacapavir against HIV-2, and Exploring the Effect of Raltegravir on the Activity of SARS-CoV-2

**DOI:** 10.3390/v16101607

**Published:** 2024-10-13

**Authors:** Irene Wanjiru Kiarie, Gyula Hoffka, Manon Laporte, Pieter Leyssen, Johan Neyts, József Tőzsér, Mohamed Mahdi

**Affiliations:** 1Laboratory of Retroviral Biochemistry, Department of Biochemistry and Molecular Biology, Faculty of Medicine, University of Debrecen, 4032 Debrecen, Hungary; kiarie.irene@med.unideb.hu (I.W.K.); hoffka.gyula@med.unideb.hu (G.H.); 2Doctoral School of Molecular Cell and Immune Biology, University of Debrecen, 4032 Debrecen, Hungary; 3Laboratory of Virology and Chemotherapy, Rega Institute for Medical Research, Department of Microbiology, Immunology and Transplantation, KU Leuven, 3000 Leuven, Belgium; manon.laporte@kuleuven.be (M.L.); pieter.leyssen@kuleuven.be (P.L.); johan.neyts@kuleuven.be (J.N.); 4European Research Infrastructure on Highly Pathogenic Agents (ERINHA-AISBL), Rue du Trône 98, 1050 Brussels, Belgium

**Keywords:** HIV-2, integrase strand transfer inhibitor (INSTIs), HIV, integrase, lenacapavir, SARS-CoV-2

## Abstract

Retroviruses perpetuate their survival by incorporating a copy of their genome into the host cell, a critical step catalyzed by the virally encoded integrase. The viral capsid plays an important role during the viral life cycle, including nuclear importation in the case of lentiviruses and integration targeting events; hence, targeting the integrase and the viral capsid is a favorable therapeutic strategy. While integrase strand transfer inhibitors (INSTIs) are recommended as first-line regimens given their high efficacy and tolerability, lenacapavir is the first capsid inhibitor and the newest addition to the HIV treatment arsenal. These inhibitors are however designed for treatment of HIV-1 infection, and their efficacy against HIV-2 remains widely understudied and inconclusive, supported only by a few limited phenotypic susceptibility studies. We therefore carried out inhibition profiling of a panel of second-generation INSTIs and lenacapavir against HIV-2 in cell culture, utilizing pseudovirion inhibition profiling assays. Our results show that the tested INSTIs and lenacapavir exerted excellent efficacy against ROD-based HIV-2 integrase. We further evaluated the efficacy of raltegravir and other INSTIs against different variants of SARS-CoV-2; however, contrary to previous in silico findings, the inhibitors did not demonstrate significant antiviral activity.

## 1. Introduction

Of the approximately 40 million people living with HIV worldwide, HIV-1 infection accounts for the majority of the cases [1]. Statistical data on the prevalence of HIV-2 remain largely unknown, with earlier estimations over a decade ago suggesting that around 2–3 million people are infected with HIV-2, a descendant of the simian immunodeficiency virus of sooty mangabeys (SIV/sm), which is endemic in West Africa [2]. Since then, a significant number of cases have been reported in Europe, most prominently in France, Portugal, Spain, and Italy [3], most probably as a result of colonial and economic ties and migration from central and Western African countries [4]. Compared to HIV-1, infection with HIV-2 is characterized by protracted latency, a lower viral load, and a higher CD4+ cell count; however, in due time, the infection may progress to the acquired immunodeficiency syndrome (AIDS) if left untreated, albeit at a much slower rate than that of its counterpart [5].

Combined antiretroviral therapy (ART) targeting different stages of the viral life cycle has revolutionized the management of HIV infection, rendering it a chronic yet manageable infection. In spite of the combination therapy, drug resistance due to continuous evolution of mutations remains the biggest drawback to the effectiveness of ART, especially in treatment-experienced patients [6]. Using a co-evolutionary probability model, protein sequence analysis has attributed this resistance to entrenchment of primary mutations induced by therapy, showing considerable variability among patient populations [7]. In addition, non-adherence and disparities in drug availability may contribute to subtype-specific or cross-resistance emergence, impacting transmission and treatment outcomes, consequently limiting treatment options [8]. This has prompted the development of long-acting oral or injectable drugs with potential for monotherapy that allow for pre-exposure prophylaxis (PrEP) and reduced intake intervals. Such recently approved drugs include cabotegravir and lenacapavir, belonging to the integrase strand transfer inhibitors (INSTIs) and capsid inhibitors (CIs) groups, respectively [9].

To date, all of the approved drugs are designed for HIV-1, and their efficacy against HIV-2 remains largely unknown. Bearing in mind the presence of inherent polymorphisms in its genetic sequence and differences in its replication dynamics, analyzing the efficacy of ART in the context of HIV-2 is indeed important. Research has shown that HIV-2 is intrinsically resistant to non-nucleoside reverse transcriptase inhibitors [10], and many protease inhibitors [11], and is widely susceptible to nucleoside reverse transcriptase inhibitors (NRTIs) and INSTIs [12].

The retroviral integrase is derived from the Gag-Pol polyprotein as a 32 kDa protein composed of three functional domains: the amino zinc binding terminal domain comprising of histidine and cysteine residues, the DNA binding carboxyl domain, and the enzyme’s active central catalytic core [13]. Integrase mediates the 3’-end processing and strand transfer, which are key steps in integration. Conversely, INSTIs compete for the enzyme active binding sites, including a divalent ion impeding the merging of viral and host cell DNA and thereby blocking integration [14]. To date, five INSTIs are approved for clinical use, classified into first-generation inhibitors: raltegravir and elvitegravir, and the second-generation ones: dolutegravir, bictegravir, and cabotegravir [15]. Initially, INSTIs exhibited potent efficacy against limited strains of HIV-2; however, this was short-lived as resistance-associated mutations emerged in raltegravir and elvitegravir-treated patients [16]. Second-generation inhibitors soon followed, with a change from bicyclic to tricyclic ring at the central pharmacore of the drugs, boosting their potency. Dolutegravir showed higher stability, tolerability, and efficacy against mutants resistant to raltegravir and elvitegravir, as reported in an in vitro study against HIV-2 in patients with resistance to first-generation inhibitors [17]. A five-amino acid insertion at the carboxy-terminal domain of HIV-2 integrase resulted in high-level resistance to raltegravir and elvitegravir and moderate resistance to dolutegravir [18,19,20]. Bictegravir showed high potency against HIV-2 in a single cycle assay, with EC_50_ between 1–5 nM, although there is no published data on its clinical use in treatment-naïve individuals with HIV-2 infection [21]. Cabotegravir, approved in 2021 for use in combination with rilpivirine as a once-daily oral tablet or as a long-acting monthly injection, was reported to have an EC_50_ of 1.2–1.7 nM against HIV-2_ROD9_ in single-cycle assays [22]. A collection of studies analyzing the efficacy of INSTIs against HIV-2 and their reported IC_50_/EC_50_ values is presented in Appendix A [20,21,22,23,24,25]. Resistance to INSTIs is often attributed to insertion and substitutional mutations in the integrated viral DNA; however, the role of unintegrated viral DNA (uDNA), particularly under INSTI treatment, warrants further exploration [26]. Persistence of the 1-LTR and 2-LTR circles, especially in non-dividing cells such as macrophages or quiescent T cells, might be implicated in the development of resistance mechanisms [27,28].

The INSTIs bind to the integrase active site, which is susceptible to point mutations; therefore, exploration of alternative binding sites on the integrase enzyme that are less prone to mutations is vital; additionally, researchers are designing dual inhibitors capable of simultaneously targeting the functions of the integrase, the reverse transcriptase, and the protease enzymes [15]. Still in their early developmental stages are the allosteric integrase inhibitors (ALLINIs) that target the allosteric pockets of integrase. They work by disrupting the interaction between integrase and the transcription co-activator lens epithelium-derived growth factor (LEDGF/p75), which is crucial for integration site targeting [29]. In an in vitro study, integration of HIV-1 was hindered when the ALLINI, BD-1, was administered, inhibiting the virus with an EC_50_ range of 0.3 to 1.5 µM [30]. LEDGF inhibitors have been considered for implementation in a block-and-lock strategy aimed at achieving HIV cure. It is indeed plausible that modifying integration site preferences through small molecules early on during treatment may reduce the functional viral reservoir, allowing for subsequent shock and kill therapies targeting the latent reservoirs [31]. Future prospects also include the development of more potent long-acting formulations, such as the recently approved lenacapavir, and nano-formulations for PrEP [32]. Improvements in the pharmacokinetics of INSTIs are being explored through a long-acting slow-effective release ART (LASER ART) approach focused on developing ultra-long-acting nano-formulations, implantable formulations, and microneedle array assays, particularly for cabotegravir and dolutegravir. Promising outcomes were observed following injections in mice, rats, and rhesus macaques, with sustained drug levels maintained for up to six months [33]. Additionally, in silico studies are ongoing to assess new compounds targeting the active site of integrase, with one potential candidate being an indole-2 carboxylic acid derivative identified as a prospective HIV-1 INSTI, among others [34]. However, there is a notable lack of research regarding the application of these advancements to HIV-2.

The novel capsid inhibitor, lenacapavir, was approved in 2022 as a long-acting injectable inhibitor streamlined for use in patients failing multiple therapy and as pre-exposure prophylaxis (PrEP). Lenacapavir targets multiple phases of early and late viral life-cycle events by blocking nuclear import of the pre-integration complex, proper capsid core formation, and production of new virions [35]. Ongoing trials in HIV-1 patients reported its efficacy at very low dosage in both treatment-experienced and naïve patients. At the time of writing this manuscript, there were no planned clinical trials for HIV-2-infected patients to assess its efficacy against HIV-2. To our knowledge, it has only been assessed in two clinical isolates from human peripheral blood mononuclear cells with an EC_50_ value of 885 pM [23].

In the midst of the coronavirus disease of 2019 (COVID-19) pandemic, HIV antiretroviral drugs became, amongst others, the focus of extensive research, given the lack of novel drugs against the severe acute respiratory syndrome-2 virus (SARS-CoV-2) at the time. Among the compounds evaluated against the SARS-CoV-2 main protease (mPro) and RNA-dependent RNA polymerase were the protease [36] and integrase inhibitors raltegravir and dolutegravir. Based on the binding energies, molecular docking studies reported raltegravir as a promising candidate for the treatment of COVID-19 [37], which prompted the need for further investigations to consolidate this finding.

The susceptibility of HIV-2 to the integrase inhibitors remains widely understudied, and a treatment protocol is yet to be established. A very limited number of studies have been performed to assay for the inhibitors against HIV-2; moreover, the majority of these studies are phenotypic assays using virions isolated from patients experiencing virological failure to ART or examined the efficacy of inhibitors in INSTIs-based regimens [38,39,40]. The lack of a standardized methodology to assay for the inhibitor’s efficacy further hinders the interpretation of the results; therefore, we set out to test the efficacy of lenacapavir and the INSTIs against ROD-based HIV-2 pseudovirions in cell culture-based inhibition profiling assays, analyzing the inhibitory potential of the inhibitors in the context of a unified integrase sequence. Furthermore, utilizing computational methods of molecular docking, we created a model structure of HIV-2 integrase and analyzed its binding with dolutegravir, bictegravir, and cabotegravir, and also analyzed the interaction between lenacapavir and HIV-2 capsid protein. Additionally, in order to verify the findings of previous in silico studies, we analyzed the efficacy of raltegravir, dolutegravir, and cabotegravir against the ancestral (Wuhan) and omicron variants of SARS-CoV-2.

## 2. Materials and Methods

### 2.1. Plasmid Vectors and Inhibitors

We utilized 2nd generation lentiviral vectors for the production of HIV-1 and HIV-2 pseudovirions. For HIV-1, we used psPAX2 as a packaging plasmid, a kind gift from Dr. D. Trono (University of Geneva Medical School, Geneva, Switzerland); a modified pWOX expressing mCherry as a transfer vector [41]; and pMDG, encoding for the envelope protein of the vesicular stomatitis virus (VSV-G). For HIV-2, the plasmids included HIV-2 CGP (a ROD-based HIV-2 protein expression vector encoding HIV-2 genes), HIV-2 CRU5SINCSW; a transfer plasmid, and pMDG plasmid [42]. HIV-2 CRU5SINCSW was modified to encode mCherry fluorescent protein under a CMV promoter in between BamHI and ECoRI restriction sites. Success of cloning and transduction efficiency were confirmed by restriction reactions, PCR, and transduction assays. HIV-2 vectors were a kind gift from Joseph P. Dougherty from the Robert Wood Johnson Medical School (NJ, USA). All the inhibitors were obtained from MedChem Express (MCE, NJ, USA) and diluted in DMSO in concentrations ranging from 1 pM to 1 µM.

### 2.2. Production of HIV-1 and HIV-2 Pseudovirions

We carried out pseudovirion production in 293T human embryonic kidney (HEK-293T) cells (Invitrogen, Carlsbad, CA, USA). On day 1, the cells were seeded in a T-75 flask in 15 mL of full DMEM containing 10% fetal bovine serum (FBS), 1% L-glutamine, and 1% penicillin-streptomycin. Cells achieved about 70% confluency (5–6 × 10^6^ cells/mL) after 24 h. The following day, transfection was carried out using polyethylenimine (PEI) (Sigma-Aldrich, St. Louis, MO, USA), and 10 µg of the plasmids and the transfected cells were incubated at 37 °C with 5% CO_2_ in 5 mL of antibiotic-free DMEM containing 1% FBS after adding the PEI-DNA solution for 5–6 h. Thereafter, the media was replaced with 10 mL of full DMEM, and the supernatant containing virions was collected after 24 h for three consecutive days and filtered through a 0.45 μm polyvinylidene fluoride filter (Merck Millipore, Darmstadt, Germany). The collected supernatant was concentrated by ultracentrifugation (100,000× *g*, 2 h, 4 °C), and the viral pellet was collected in 200 μL phosphate-buffered saline (PBS), aliquoted, and stored at −70 °C. An ELISA-based colorimetric reverse transcriptase (RT) assay to measure the RT activity of the viruses (Roche Applied Science, Mannheim) was then used to measure the concentration of the pseudovirions in nanograms of reverse transcriptase (RT)/well unit.

### 2.3. Pseudovirion Production to Assay for Effects of Lenacapavir on HIV-1 and HIV-2 Infectivity Using Pseudovirions with Aberrant Capsid

Using the same protocol as above, viruses were produced in the presence of different concentrations of lenacapavir before transduction experiments. On day 1, three million HEK-293T cells were seeded in T-75 flasks in 10 mL full DMEM. On day 2, the cells were treated for 3 h with 1 nM, 10 nM, and 100 nM lenacapavir and thereafter transfected with 10 µg of HIV-1 and HIV-2 vectors in 5 mL of 1% DMEM. After 5–6 h of incubation, medium was changed with fresh full DMEM supplemented with the same concentrations of lenacapavir, followed by incubation for two days. The virion-containing supernatant was then collected and concentrated with Amicon Ultracel 100K (Merck Millipore Ltd., Tullagreen, Ireland), and an ELISA-based colorimetric RT assay was used to measure the concentration of virions, followed by transduction assays.

### 2.4. MTT Cell Proliferation Assay

96-well plates were seeded with 25,000 HEK-293T cells in full DMEM (10% FBS, 1% L-glutamine, 1% penicillin/streptomycin). The following day, cells were treated with a serial dilution of the INSTIs ranging from 1 nM to 1 µM and 100 pM to 100 nM of lenacapavir and incubated for two days. The MTT (Thermo Fisher Scientific, Waltham, MA, USA) assay was then carried out to assess the viability of the treated cells following the manufacturer’s protocol.

### 2.5. Inhibition Profiling in Jurkat Cells

48-well plates were seeded with 25,000 Jurkat cells (ATCC, Manassas, VA, USA) in 200 µL of RPMI medium complemented with 10% FBS and 1% L-glutamine for 24 h. Cells were then treated with a serial dilution of the inhibitors ranging from 1 nM to 1 µM of INSTIs (100 pM and 500 pM additional concentrations were used for some inhibitors to better calculate IC_50_ values) and 100 pM to 100 nM of lenacapavir in fresh medium without antibiotics and incubated at 37 °C for 3 h; thereafter, the cells were transduced with 4 ng equivalent to RT/well activity of HIV-1 or HIV-2 in the presence of 8 µg/mL polybrene/well and further incubated at 37 °C for 48 h. Cells were collected in 400 µL cold sterile PBS, and quantitative analysis of the transduction efficiency in the presence of the inhibitors was assessed by flow cytometry (FACS caliber, BD Biosciences, Franklin Lakes, NJ, USA). The results were analyzed by FlowJo Software Version 10 (Becton, Dickinson, and Company, 2019) to determine the percentage of mCherry fluorescence in 5000 cells, indicating transduction by HIV-2 and HIV-1, respectively. GraphPad Prism ver. 9 (GraphPad Software, Inc. Boston, MA, USA) performed the IC_50_ calculations and graphs.

### 2.6. Inhibition Assay of INSTIs against SARS-CoV-2

VeroE6-GFP cells (provided by M. van Loock, Janssen Pharmaceutica, Beerse, Belgium) were seeded at a density of 25,000 cells/well in 96-well plates (Greiner Bio One, catalog no. 655090) and pre-treated with three-fold serial dilutions of the compounds overnight in presence of the MDR1-inhibitor CP-100356 (final concentration 0.5 μM). On the next day (day 0), cells were infected with the SARS-CoV-2 inoculum at a multiplicity of infection (MOI) of 0.001 median tissue culture infectious dose (TCID50) per cell. The number of fluorescent pixels of the GFP signal determined by High-Content Imaging (HCI) on day 4 post-infection (p.i.) was used as a read-out. Percentage of inhibition was calculated by subtracting background (number of fluorescent pixels in the untreated-infected control wells) and normalizing to the untreated-uninfected control wells (also background subtracted). The 50% effective concentration (EC_50_, the concentration of compound required for fifty percent recovery of cell-induced fluorescence) was determined using logarithmic interpolation. A similar protocol was used to determine antiviral activity in A549-Dual™ hACE2-TMPRSS2 cells, but no MDR1-inhibitor CP-100356 was used, and the cell viability was determined 4 days p.i. using viability staining with MTS (3-(4,5-dimethylthiazol-2-yl)-5-(3-carboxymethoxyphenyl)-2-(4-sulfophenyl)-2H-tetrazolium). The percentage of antiviral activity was calculated by subtracting the background and normalizing to the untreated-uninfected control wells, and the EC_50_ was determined using logarithmic interpolation. In both cell lines, potential toxicity of compounds was assessed in a similar set-up in treated-uninfected cultures where metabolic activity was quantified at day 5 using the MTS assay as described earlier [43]. The 50% cytotoxic concentration (CC_50_, the concentration at which cell viability reduced to 50%) was calculated by logarithmic interpolation.

SARS-CoV-2 GHB (EPI ISL407976|2020-02-03) was recovered from a nasopharyngeal swab taken from an asymptomatic patient returning from Wuhan, China. Virus stocks were inoculated and passaged first in HuH-7 cells and then passaged 7 times on VeroE6-eGFP cells. GHB-03021 has a ΔTQTNS deletion at 676–680 residues that is typical for SARS-CoV-2 strains that have been passaged several times on VeroE6 cells [44].

SARS-CoV-2 variants B.1.1.7 (GISAID ID: EPI_ISL_791333), Omicron BA.2 (GISAID ID: EPI_ISL_10654979), and Omicron BA.5 (GISAID ID: EPI_ISL_14782497) were recovered from nasopharyngeal swabs of RT-qPCR-confirmed human cases. Virus stocks were generated by passaging the virus in Calu-3 cells, followed by the production of a screening virus stock on A549+hACE2+hTMPRSS2 cells.

All SARS-CoV-2 manipulations were performed in biosafety level 3 (BSL-3) and 3+ (CAPs-IT) facilities at the Rega Institute for Medical Research, KU Leuven, according to institutional guidelines.

### 2.7. Molecular Docking of Integrase Inhibitors

We modeled HIV-2 integrase structure based on the available Simian immunodeficiency virus red-capped mangabeys (SIVrcm) intasome structure in complex with bictegravir and DNA substrate using SWISS-MODEL (PDB ID: 6RWM) [45]. The original SIV DNA sequence was modified to represent our available HIV-2 sequence. Nucleotide transmutation was carried out with Chimera [46], which was also applied to examine the histidine protonation states. The protonation states of titratable residues were evaluated with PROPKA [47,48]. The structure was minimized with the Amber16 software [49,50,51], with ff14SB [52] and DNA OL15 [53] force fields applied, also applying the Li/Merz ion parameters (12-6-4 set) for the divalent Mg^2+^ ions [54], along with solvation in TIP3P [55] water molecules. Ions were added to approximate a 0.15 M NaCl concentration, modified in accordance with the charge split protocol [56]. Bictegravir and cabotegravir inhibitors were optimized with Gaussian16 software [57], applying the M06-2X [58] method and the 6-311+G (d, p) basis set for optimization, followed by RESP [59] charge calculation with antechamber [60]. Docking was carried out with the PLANTS [61,62] software to subunit A of the integrase complex, which also included the bound DNA strands, the Zn^2+^ ions, the Mg^2+^ ions, and three water molecules coordinated by the Mg^2+^ ions. The center of the docking was defined as the geometric point halfway between the Mg^2+^ ions. The binding site radius was 12 Å. The structure with the lowest energy was accepted.

### 2.8. Molecular Docking of Lenacapavir

We used the HIV-2 capsid protein (p26) sequence (UniProt: P04590) to generate a model structure of the protein, applying AlphaFold [63] through ColabFold [64]. We converted both the HIV-2 capsid model and the lenacapavir pdb model to pdbqt files with AutoDock Tools. Docking was carried out with AutoDock Vina 1.1.2 [65], centered on the OD1 atom of Asn56 of the HIV-2 model. Lenacapavir is bound to the proximity of the Asn57 in the structure of the HIV-1 capsid protein (p24) structure that binds lenacapavir (PDB ID: 7RHN [66]). The 7RHN structure was used for comparison.

## 3. Results

### 3.1. Susceptibility of HIV-2 to INSTIs

Using a standard protocol, we confirmed that all the integrase inhibitors were effective against ROD-based HIV-2 integrase at low nanomolar concentrations (Table 1 and Figure 1), similar to HIV-1, and as reported elsewhere in the literature for some HIV-2 isolates in clinical and in vitro cell culture studies [16,17,18,19,20,21,22,60,65].

### 3.2. In Silico Analysis of Integrase Inhibitors

We designed an HIV-2 integrase model based on SIV integrase. To examine the quality of our model, we have compared the Ramachandran plots of the SIV structure with our minimized model structure (monomer A), applying the Ramachandran plot function of Chimera [38] (Appendix A). Using the PLANTS molecular docking software, we successfully docked bictegravir, cabotegravir, and dolutegravir at the integrase active sites. Visual comparison of the docked dolutegravir conformation with the dolutegravir bound in the SIV intasome (PDB ID: 6RWM) showed significant resemblance, confirming the possibility of inhibitor binding at the active site.

Results of the docking are visualized in Figure 2. A tight binding of the inhibitor to the active site was observable in the case of all of the inhibitors.

### 3.3. Efficacy of Lenacapavir against HIV-2

We evaluated the inhibitory effect of lenacapavir on HIV-2 transduction efficiency and virion formation. Following treatment of cells with the inhibitor, we observed that lenacapavir inhibited pseudovirion transduction in picomolar concentration, as previously shown for HIV-1 [29]. To assess the inhibitory effects of lenacapavir against the formation of HIV-2 pseudovirions, we produced the virions in the presence of various concentrations of lenacapavir (1, 10, and 100 nM). A RT ELISA-based colorimetric assay was carried out on the produced virions to assess for activity, and we did not detect a significant difference in RT activity between HIV-2 virions produced in the presence of lenacapavir and that of the control. However, when we transduced cells with pseudovirions produced in the presence of lenacapavir, transduction efficiency was severely compromised (Figure 3), indicating that the virions were defective despite having normal RT activity (Appendix A). To examine the molecular background of lenacapavir binding to capsid protein, we have performed a docking simulation of lenacapavir to the HIV-2 capsid protein (p26) model, visualizing the energetically most preferable docked conformer (Figure 4).

### 3.4. Assessment of Anti-SARS-CoV-2 Activity of Raltegravir

Inhibition profiling of raltegravir in Vero E6 cells and A549 cells against SARS-CoV-2 showed that the inhibitor did not exert significant antiviral activity against neither the prototypical Wuhan-Hu-1 (GBH) nor the B.1.1.7 and omicron BA.2 variants (IC_50_ > 100 μM) (Figure 5). It is noteworthy that we also evaluated the efficacy of dolutegravir and cabotegravir against SARS-CoV-2 utilizing the same methodology; similarly, no significant inhibition was observed in our analyses.

## 4. Discussion

In our study, we conducted cell culture-based inhibition profiling assays of INSTIs against ROD-based HIV-2 pseudovirions, comparing it to HIV-1. Our results show that all of the commercially available INSTIs exhibited efficacy against HIV-2 in nanomolar concentrations. Raltegravir inhibited HIV-2 integrase with an IC_50_ value of 2.1 nM, comparable to an in vitro phenotypic study conducted soon after its approval against HIV-2 ROD clinical isolates [67]. Several clinical trials have demonstrated the effectiveness of raltegravir in suppressing viral load in treatment-naïve and experienced HIV-2 infected patients, although the N155H resistance mutation quickly emerged [46]. Elvitegravir was shown to have an IC_50_ of 0.7 nM [67], comparable to a cell culture study by Zheng et al. that reported elvitegravir to have an IC_50_ ranging from 0.3–0.9 nM in clinical isolates from integrase-naïve patients [24]. In our study, elvitegravir inhibited HIV-2 with an IC_50_ of 2.1 nM, which was slightly higher but nonetheless comparable.

Dolutegravir was reported to be an active integrase inhibitor in the treatment-naïve patients [17], and similarly, an observational study in India showed that INSTIs-naïve HIV-2 patients on dolutegravir regimen had undetectable viral load [19]. In our study, dolutegravir showed inhibition with an IC_50_ of 0.9 nM. Bictegravir, cabotegravir, and lenacapavir are the latest approved inhibitors, and there is limited data on their efficacy against HIV-2. Based on dolutegravir, bictegravir was shown to be very potent at inhibiting HIV-1, resulting in a low rate of resistance mutations compared to raltegravir, elvitegravir, and dolutegravir [24]. In a recently published in vitro study, HIV-2 was sensitive to cabotegravir with a low EC_50_ of 1.8–2.6 nM in a single-cycle spreading infection assay [22]. A study by Hingrat et al. reported bictegravir to be more superior to cabotegravir [18]. In our study, bictegravir and cabotegravir were both effective against HIV-2 ROD-based pseudovirions with an IC_50_ of 1.8 nM and 2 nM, respectively.

Since the COVID-19 pandemic, many studies assessed the repurposing of antiretroviral drugs against various SARS-CoV-2 variants, yielding mixed results. An in vitro and in silico investigation analyzed six antiretroviral drugs, including raltegravir, in Vero E6 cells against the D614G variant from a Colombian isolate. Raltegravir demonstrated inhibitory effects on the strain at concentrations of 6.3–25 µM and also exhibited the highest binding energies in the in silico analysis [68]. On the other hand, an in vivo study revealed that INSTIs and tenofovir did not influence the acquisition or outcomes of SARS-CoV-2 infection among a Dutch cohort of individuals living with HIV [69].

As the only one of its class, lenacapavir is the latest addition to the arsenal in the fight against HIV, inhibiting the early phase of viral life cycle and capsid formation [70]. Lenacapavir was effective against HIV-1 in our study with an IC_50_ of 399.3 pM, slightly higher than results of cell culture studies in HIV-1, which reported its potency in the picomolar range of (EC_50_ 50–314 pM) [71]. In vitro cell culture studies by Link O. et al. reported an EC_50_ of 105 pM against HIV-1 in MT-4 cells, 56 pM in macrophages, 32 pM in primary CD4 cells, and in 23 clinical HIV-1 isolates ranging from 20–160 pM in human peripheral blood mononuclear cells. In addition, this study also reported efficacy of the inhibitor against two HIV-2 isolates [23]. At the time of writing this manuscript, that was the only study evaluating the efficacy of lenacapavir against HIV-2. In our study, lenacapavir inhibited HIV-2 with an IC_50_ of 206.2 pM, within the efficacy range to that exhibited by HIV-1 [62]. Lenacapavir exerts its antiviral effects by targeting the interface between two capsid hexamer subunits, thereby disrupting HIV replication [72]. This interaction leads to the production of virions with structurally compromised capsids, which are capable of infecting new target cells but are unable to undergo further replication [73]. Our findings corroborated this mechanism, demonstrating that pseudovirions generated in the presence of lenacapavir exhibited a marked deficiency in transduction efficiency in cells infected with either HIV-1 or 2.

Molecular docking of bictegravir, cabotegravir, and dolutegravir to our HIV-2 integrase model provides theoretical evidence of the inhibitors binding to the enzyme. Structural analysis suggests that the active site Mg^2+^ ions have a crucial role in inhibitor binding, forming interactions that were described as almost covalent between bictegravir and the SIVrcm integrase [67]. In that study, resistance mutations of SIVrcm integrase were shown to modulate the charge distribution of the Mg^2+^ ions, raising the possibility of similar effects in HIV-2 as well as HIV-1 integrase. Given the lack of studies on resistance mutations in the case of HIV-2 integrase, it would be important to analyze the effects of the mutations in the context of this coordinated binding to the Mg^2+^ ions in future studies. Molecular docking of lenacapavir to the AlphaFold model of HIV-2 capsid protein (p26) resulted in a bound inhibitor conformer.

## 5. Conclusions

To our knowledge, no study has evaluated the efficacy of all of the available INSTIs against HIV-2 integrase. Our analysis is perhaps advantageous because we analyzed the efficacy of the inhibitors against the same ROD-based integrase sequence, utilizing a standardized cell culture-based methodology, without influence or interference from other drugs. Only a few studies have reported on the efficacy of bictegravir and cabotegravir against HIV-2, and in regards to lenacapavir, it was only evaluated against two isolates. On a separate note, we experimentally verified that raltegravir, contrary to what previous in silico studies had hoped for, did not exert significant antiviral activity against SARS-CoV-2; in fact, other INSTIs such as cabotegravir and dolutegravir were also found to be ineffective. This highlights the importance of in vitro and cell culture-based inhibition profiling studies in verifying in silico findings.

## Figures and Tables

**Figure 1 viruses-16-01607-f001:**
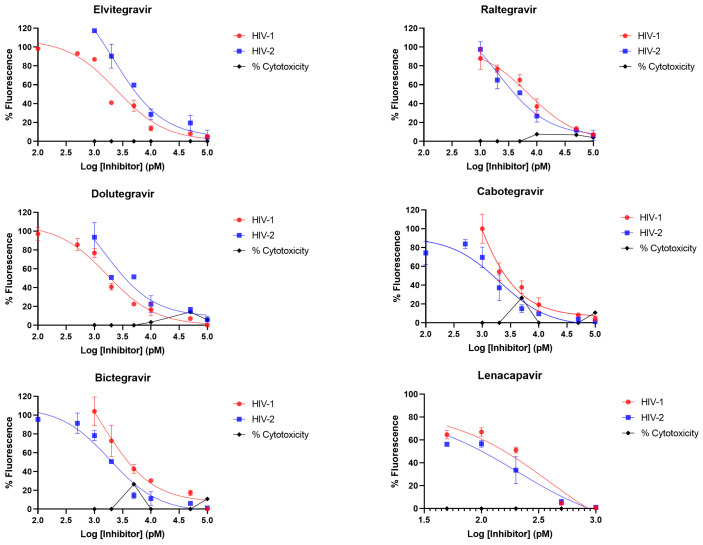
Inhibition profiling in cell culture. Four-parameter dose-response curves are displayed. *y* axis indicates percentage of fluorescence (infectivity) compared to negative control (cell treated with DMSO without the inhibitor); *x* axis is the logarithmic transformation of the inhibitor’s concentration. Percentage of cytotoxicity is also indicated, which can be interpreted from the *y* axis compared to the control. Results are concluded from triplicate measurements.

**Figure 2 viruses-16-01607-f002:**
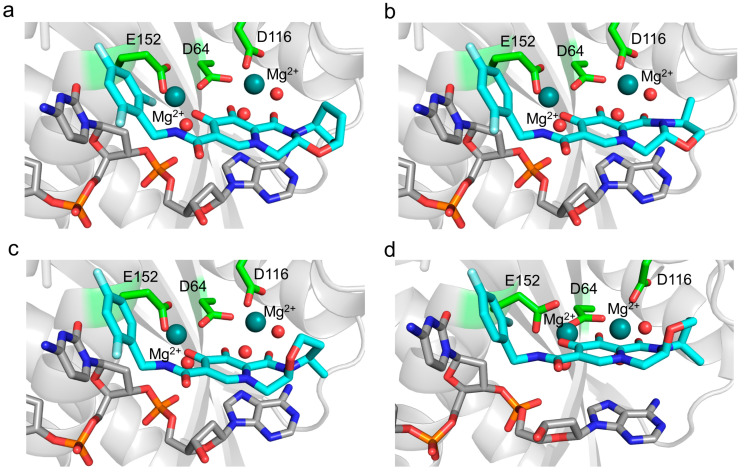
Docked integrase inhibitors compared to experimental structure. Structure of (**a**) bictegravir, (**b**) cabotegravir, and (**c**) dolutegravir inhibitors docked by PLANTS in the active site of HIV-2 integrase. The inhibitor (teal), the DNA substrate (gray), and the Mg^2+^ coordinating active site residues (green) are represented in stick format, the Mg^2+^ ions (deep teal), and the coordinated water molecules (red) as spheres. (**d**) For comparison, the SIV integrase crystal structure (PDB ID: 6RWN [67]) that binds dolutegravir, applying the same coloring and representation schemes.

**Figure 3 viruses-16-01607-f003:**
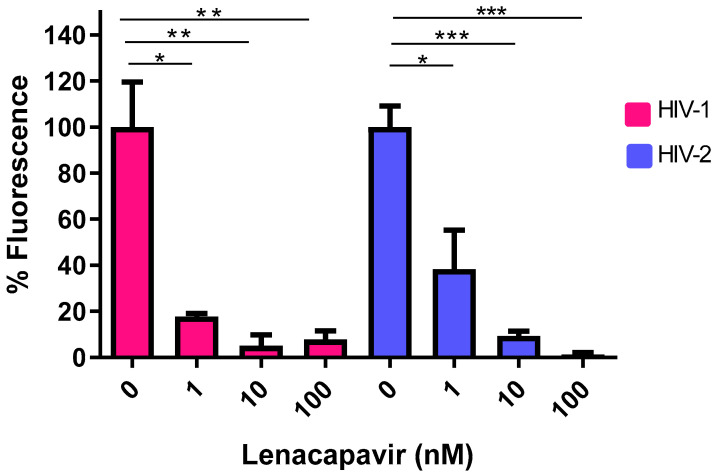
HIV-1 and 2 pseudovirions with aberrant capsids produced in the presence of lenacapavir. Pseudovirions were produced in the presence of 1, 10, and 100 nM of lenacapavir. The y axis indicates percentage of transduction efficiency (percentage of fluorescence), and x axis denotes the concentrations of lenacapavir used to treat the cells. Control cells were only treated with DMSO. Results are concluded from triplicate measurements. * *p* value ≤ 0.05, ** *p* value ≤ 0.01, *** *p* value ≤ 0.001.

**Figure 4 viruses-16-01607-f004:**
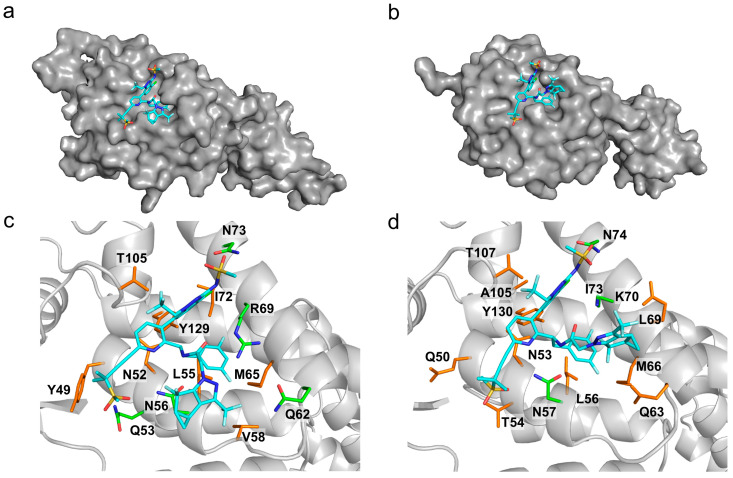
(**a**) Lenacapavir in complex with the HIV-2 capsid protein (p26) docked by AutoDock Vina to the AlphaFold model; (**b**) in complex with the active site of HIV-1 capsid protein (p24), crystal structure (PDB ID: 6V2F). The protein is represented with a gray surface and lenacapavir as blue sticks. Closeup of lenacapavir in (**c**) the HIV-2 model and (**d**) the HIV-1 crystal structure. The inhibitor (teal) and the interacting capsid residues (hydrogen bond forming—green, apolar interaction forming—brown) are represented as sticks, the rest of the protein as cartoon. The interactions were determined with LigPlot+.

**Figure 5 viruses-16-01607-f005:**
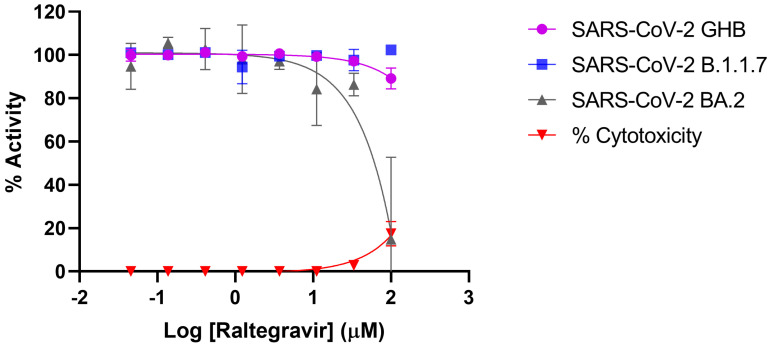
Inhibition profiling of Raltegravir against SARS-CoV-2. Efficacy against the prototypical Wuhan-Hu-1 (GBH) is shown in purple, while that against the B.1.1.7 variant is shown in blue, and omicron BA.2 is shown in grey. Percentage of cytotoxicity is indicated in red. Inhibition profiling was carried out in Vero E6 cells and then confirmed in A549-Dual™ hACE2-TMPRSS2 cells. *y* axis indicates percentage of inhibition, and *x* axis is the logarithmic transformation of the inhibitor’s concentration. Results represent at least triplicate measurements.

**Table 1 viruses-16-01607-t001:** Results of inhibition profiling in cell culture. ± indicate standard error (SE) displayed as logIC_50_ value of the inhibitor. Results were concluded from triplicate measurements.

	IC_50_ (nM)	
**INSTI**	**HIV-1**	**HIV-2**
Elvitegravir	2.4 (±0.08)	2.1 (±0.1)
Raltegravir	7.2 (±0.1)	2.1 (±0.1)
Dolutegravir	2.2 (±0.07)	0.9 (±0.4)
Bictegravir	1 (±0.3)	1.8 (±0.08)
Cabotegravir	0.4 (±0.3)	2 (±0.1)
**Capsid Inhibitor**	**IC_50_ (pM)**	
Lenacapavir	399.3 (±0.2)	206.2 (±0.2)

## Data Availability

All required data are available in the manuscript. Any additional data can be provided upon request.

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
