# Peer review of "Efficacy of Integrase Strand Transfer Inhibitors and the Capsid Inhibitor Lenacapavir against HIV-2, and Exploring the Effect of Raltegravir on the Activity of SARS-CoV-2"

_viruses, 2024, doi:10.3390/v16101607_

Round 1

Reviewer 1 Report (Previous Reviewer 1)

Comments and Suggestions for Authors

The authors addressed my concerns, and I recommend this manuscript for publication.

Author Response

Comment: The authors addressed my concerns, and I recommend this manuscript for publication.

Reply: Dear Reviewer, thank you again for your recommendation and the time spent revieweing and helping improving this manuscript.

Reviewer 2 Report (Previous Reviewer 2)

Comments and Suggestions for Authors

Kiarie & co-authors provide a revised version of their manuscript, describing an in vitro analysis of the efficacy of INSTIs and lenacapavir against an HIV-2 prototype strain, and one INSTI molecule for its activity against SARS-CoV2 replication in vitro.

The revised version of the manuscript is much better on many points, and I thank the authors for their efforts.

However, it remains one point which was not correctly addressed:

“(...)the calculation of IC50 (table 1) is not supported by the dose-response curves of figure 1. (...), these dose-response curves are incomplete, meaning that the associated IC50 is at best an estimation (...)”

To answer this point, the authors reanalyzed the data with a new software and changed the display of the figures.

However, this does not solve the problem, which is that the range of concentrations tested is incomplete. For example, the IC50 of the cabotegravir against HIV-2 (0.8nM) is lower than the lowest concentration tested (1nM). This is evident from figure 1, where the % of fluorescence at log(concentration)nM=0 (hence concentration=1nM) is less than 50%, meaning that cabotegravir inhibits HIV-2 for more than 50%. This demonstrates that the lowest compound concentration tested is still higher than the concentration that inhibits 50% of the effect, which is the definition of IC50. How can an IC50 value be accurately calculated when it is never reached? Therefore, these experiments must include a broader range of low concentrations, e.g. from 0.001nM to 1µM, to ensure that this range encompasses the IC50 concentration, which is a necessary condition for an accurate calculation of the IC50 value. The same holds for the other compounds where the IC50 is lower than the lowest concentration tested (dolutegravir and bictegravir).

 Other points:

·       If the authors are keeping their results on Sars-Cov2, then this should appear in the manuscript title.

·       Figure 1: please explain how the log scale can go from 0 to 2 Log [inhibitor] in nM : if the maximum concentration tested was 1µM, then the maximum X value should be 3 (log (1000)), not 2.

·       Same question concerning the X-axis of the lenacapavir panel. If the concentrations tested are from 1 to 1000pM, then the axis should go from 0 (log (1)) to 3, and not from 1.8 to 3.

Author Response

Comment: 

Kiarie & co-authors provide a revised version of their manuscript, describing an in vitro analysis of the efficacy of INSTIs and lenacapavir against an HIV-2 prototype strain, and one INSTI molecule for its activity against SARS-CoV2 replication in vitro.

The revised version of the manuscript is much better on many points, and I thank the authors for their efforts.

However, it remains one point which was not correctly addressed:

“(...)the calculation of IC50 (table 1) is not supported by the dose-response curves of figure 1. (...), these dose-response curves are incomplete, meaning that the associated IC50 is at best an estimation (...)”

To answer this point, the authors reanalyzed the data with a new software and changed the display of the figures.

However, this does not solve the problem, which is that the range of concentrations tested is incomplete. For example, the IC50 of the cabotegravir against HIV-2 (0.8nM) is lower than the lowest concentration tested (1nM). This is evident from figure 1, where the % of fluorescence at log(concentration)nM=0 (hence concentration=1nM) is less than 50%, meaning that cabotegravir inhibits HIV-2 for more than 50%. This demonstrates that the lowest compound concentration tested is still higher than the concentration that inhibits 50% of the effect, which is the definition of IC50. How can an IC50 value be accurately calculated when it is never reached? Therefore, these experiments must include a broader range of low concentrations, e.g. from 0.001nM to 1µM, to ensure that this range encompasses the IC50 concentration, which is a necessary condition for an accurate calculation of the IC50 value. The same holds for the other compounds where the IC50 is lower than the lowest concentration tested (dolutegravir and bictegravir).

Reply: 

  • Thank you for your insightful feedback. We would like to address your concern regarding the range of concentrations tested in our IC50 determination. Our concentration range was selected based on both prior studies and preliminary data, which suggested that this range may adequately capture the inhibition dynamics of the inhibitors against HIV-2. The IC50 as previously described is calculated through non-linear regression, fitting a sigmoidal dose-response curve to the data, and the IC50 value is reliably extrapolated from the inhibition trend observed at higher concentrations. The software used (GraphPad Prism) ensures that IC50 values are derived even when not all points fall exactly at the 50% inhibition mark. Nevertheless, as requested by the reviewer, we carried out additional experiments and added additional points below 1 nM (100, and 500 pM), in the case of inhibitors that had an IC50 below 1 nM. The figure has now been updated to include newer data points for the said inhibitors, and the x-axis was changed to the logarithmic transformation of pM concentrations. The methodology was also updated accordingly. While some values didn’t change (dolutegravir), minor change was observed for elvitegravir, bictegravir and cabotegravir). All the new values and changes to the manuscript are highlighted in red.

While we acknowledge the small change in single-digit nanomolar concentrations, we believe it is not biologically significant in the context of this study. Given the robust inhibitory effects observed, this minor variation does not alter the overall conclusion regarding the efficacy of the inhibitors against HIV-2. Furthermore, biological variability at such concentrations is unlikely to impact the relevance of the findings presented in this manuscript.

Comment: If the authors are keeping their results on Sars-Cov2, then this should appear in the manuscript title.

Reply: 

  • As requested, the title has now been modified to “Efficacy of Integrase Strand Transfer Inhibitors and the capsid inhibitor Lenacapavir against HIV-2, and exploring the effect of Raltegravir on the activity of SARS-CoV-2”

Comment: Figure 1: please explain how the log scale can go from 0 to 2 Log [inhibitor] in nM : if the maximum concentration tested was 1µM, then the maximum X value should be 3 (log (1000)), not 2.

Reply:

  •    The x-axis represented the log10 of the concentration of the inhibitor in nM. The concentration range tested in our experiments spanned from 1 nM to 1 µM (with some inhibitors now tested with additional sub-nanomolar concentrations as described earlier). We also changed the x-axis to represent the logarithmic transformation of inhibitors concentrations in pM. While plotting the data, we used 100 nM as a maximum value, as there was no point plotting further points due to total inhibition.

Comment: Same question concerning the X-axis of the lenacapavir panel. If the concentrations tested are from 1 to 1000pM, then the axis should go from 0 (log (1)) to 3, and not from 1.8 to 3.

Reply:

-    The concentration used for plotting the graph were from 0 – 1000 pM (0, 50, 100..etc). Since the logarithm of 0 is undefined, the next data point (50) when converted to logarithmic scale is approximately 1.69, hence this is the first point plotted in the graph.

Reviewer 3 Report (Previous Reviewer 3)

Comments and Suggestions for Authors

The authors have successfully addressed most of the reviewers' comments. However, citations of previous and related work placing the current work in the context of the field can be further improved.

Author Response

Comment: The authors have successfully addressed most of the reviewers' comments. However, citations of previous and related work placing the current work in the context of the field can be further improved.

Reply:

  • Dear Reviewer, thank you once more for your time and effort reviewing and helping improving this manuscript. We have added more literature data in the introduction, all the new additions and references are highlighted 

Round 2

Reviewer 2 Report (Previous Reviewer 2)

Comments and Suggestions for Authors

Thank you for these modifications and clarifications.

This manuscript is a resubmission of an earlier submission. The following is a list of the peer review reports and author responses from that submission.

Round 1

Reviewer 1 Report

Comments and Suggestions for Authors

Title: Efficacy of Integrase Strand Transfer Inhibitors and the capsid inhibitor Lenacapavir against HIV-2

Target Journal: Viruses

Manuscript ID: viruses-2929854

Manuscript Type: Article

Comments:

HIV-1 integrase (IN) is one of the key enzymes in the viral life cycle and it catalyze the integration of the viral genome into the host genome. Integrase strand transfer inhibitors (INSTIs) including Raltegravir (RAL), Elvitegravir (EVG), Dolutegravir (DTG), Bictegravir (BIC), Cabotegravir (CAB) have been approved by FDA for the anti-AIDS treatment. The manuscript reports the inhibition profile of FDA-approved INSTIs and capsid inhibitor lenacapavir against HIV-2 in cell utilizing pseudovirion inhibition profiling assays. However, the part of RAL against the SARS-CoV-2 Omicron variant in antiviral assays does not match the main topic about HIV-2 inhibition. Publish is recommended after revision.

The manuscript is also confused by the following points, which should be fixed and simplified.

·         There are not enough data points in the curve of Lenacapavir in Figure 1, which might lead to inaccurate result.

·         More data points should be included in the inhibition curves in Figure 4 to make it more accurate.

·         The terms are not consistent. “SARs-CoV-2” should be “SARS-CoV-2”, “Mg2+” should be “Mg2+” et. al.

·         The acronyms abbreviations have not been defined upon the first use for examples “pre-exposure prophylaxis” should be “pre-exposure prophylaxis (PrEP)” in line 48, page 2; and so on. These abbreviations should be used after the definition but not the full term. “pre-exposure prophylaxis (PrEP)” should be “PrEP” in line 83, page 2.

·         The typo “DAN” should be changed to “DNA” in line 219 in page 5.

Reviewer 2 Report

Comments and Suggestions for Authors

Kiarie & co-authors describe an in vitro analysis of the efficacy of molecules that are used against HIV-1 infection against an HIV-2 prototype strain. Namely, the authors have focused on integrase strand transfer inhibitors (INSTIs) and lenacapavir, a recently described drug targeting HIV-1 capsid, and have tested these inhibitors for their efficacy to inhibit the transduction by HIV-2 based pseudoviruses in Jurkat cells. For some INSTI, the author provide a molecular model of the interaction of the compounds with a structural model of HIV-2 integrase. Finally, they tested a one INSTI molecule for its activity against SARS-CoV2 replication in vitro.

This aim of paper is important, as it has been described that some drugs were active against HIV-1 but not HIV-2. Thus, exploring whether recently developed anti-HIV-1 molecules are active against HIV-2 is pertinent as HIV-2 remains an important health issue.

Unfortunately, the paper has several flaws that have to be addressed before the paper can be considered for publication:

1-      The Sars-CoV2 part is not relevant nor conclusive and must be deleted: it has nothing to do with the scope of the paper nor the special issue. Moreover, it is not conclusive as the inhibition is detected when cytotoxicity appears. Furthermore, only one of the two cellular model is displayed. Finally, there is no way to calculate an IC50 from this curve as the maximal plateau is not reached.

2-      This is in fact a major flaw in this paper: the calculation of IC50 (table 1) is not supported by the dose-response curves of figure 1. Indeed, the IC50 calculation is based on the minimal and maximum plateaus from a sigmoid-like curve, which are lacking for all the curves. Even if some of these plateaus might be sometimes extrapolated, these dose-response curves are incomplete, meaning that the associated IC50 is at best an estimation, but could be wrong of several orders of magnitude given the fact that the x-axis is logarithmic.

There is also a problem with the results displayed in table 1: indeed, if the IC50 is in nM, the SE should also be in nM and not, as stated by the authors, as “LogIC50 value”. As it is, if SE=0.3 is a log value, it means that it is a SE +/ of 2 nM. The efficacy of a compound with an IC50 of 1.8+/-2nM (bictegravir) becomes then highly questionable, especially in view of doubts about the calculation of the IC50 and the absence of complete inhibition curve.

Altogether, the authors must provide complete dose-response curves and nM SE values before being able to draw any conclusion on the efficacy of the compounds in their experimental settings. These data are necessary to make the results of the paper convincing.

In the same line, it is not realistic to display a dose response curve with only four points (lenacapavir). Please provide more points for this inhibition curve to convince the reader of the reality of the two plateaus and thus of the observed IC50.

 3-      Figure 2: the in silico modeling of INSTI binding is performed against a model of HIV-2 structure. Such sequential modelling could go wrong very quickly if one or both of the models are not correct. Thus, data concerning the accuracy of the HIV-2 model (QmeanDisCo score, Ramachandran plot) must be provided (at least in supplementary data) to assess the quality of the model. Similarly, the statistic of the docking (predicted binding energy, number of poses that were evaluated) must be provided to assess the quality of the docking.

Other major points.

-          The authors repetitively state that they demonstrate that the integrase inhibitors were effective “against ROD-based HIV-2 integrase” (lines 25 and 238). This is not true: the authors are studying inhibition of HIV-2 pseudovirion in a cellular model.  Given the low sequence identity between HIV-1 and HIV-2 and in the absence of integrase-specific experiments (affinity measurements, tests on integration defective viruses…), one cannot exclude that these inhibitors are targeting another viral step in the authors’ experimental system. This is particularly true now that a role of IN in non-integration steps of the replication has been demonstrated for HIV-1 (facilitation of Tat binding to viral RNA, role in the correct packaging of viral RNA in viral capsid during virion assembly, …). This should be discussed.

-          Please do not combine green and red for curves and histograms, as these are not distinguishable by color blind viewers (especially figure 3, see pdf version of my comments for a simulation).

-          Figure 3: please explain the statistics behind the *, ** and ***  (which p value, which statistical analysis?).

Minor points

-          there are a lot of typos. A careful editing is necessary before resubmission.

-          a reference is missing lines 239-240.

-          figure 1 and 4 display a % of cytotoxicity for which no scale is given. Please clarify.

-          Lines 101-103. Please specify that it concerns integrase inhibitors.

-          Line 168: Jurkat cells are in suspension. Therefore, there is no way to achieve “50-70% confluency” which is typical from adherent cells. Please clarify.

-          Line 173: please show, at least as supp data, raw data of FACS results for one experiment with a serial dilution of at least one of the compounds. This will help assessing the quality of the data.

-          Figure 2: it is not clear which spheres are Mg2+ ions and which are water molecules. Please clarify in the caption.

-          Figure 2:  The authors must explain why three residues are identified. If they are important for the coordination of the Mg ions, please state whether or not they are conserved between

-          Figure 2: the authors could add to this figure (or as Supp data) a comparison with the experimental structure of the original SIVrcm intasome with bictegravir that was used for modelling, as it could help to validate their HIV-2 modeling strategies. In the same line, the structure of the SIV intasome with dolutegravir is also available (PDB ID 6RWN): providing a docking of this compound on the HIV-2 model and comparing it with the experimental data on SIVrcm could strengthen the in silico part of the paper.

-          HIV-2 and SIVrcm as this could have consequences on the interpretation of the data.

-          Line 80, calbotegravir displays an EC50 of 0.12 against HIV-2 isolates in a published study. Supplementary table 1: EC50 is 1.4 to 5.6nM. Please clarify.

-          What is the sequence identity between HIV-1 and HIV-2, especially for the INSTI and the lenacapavir binding pockets?  This should be discussed, as it will have consequences on possible difference of susceptibility to inhibitors.

Reviewer 3 Report

Comments and Suggestions for Authors

While the study is comprehensive, I believe the manuscript can benefit from a clearer exposition of the main advancements that are being made in the work and not available from prior literature, especially in the Introduction. Overall, the work is highly relevant, and strongly merits publication and I believe, would be of significant interest to the readership of Viruses. I found the manuscript to be well written, and only have a few comments which the authors can address to increase the ease of readability:

Major comments:

1.     While the study is comprehensive, I believe the manuscript can benefit from a clearer exposition of the main advancements that are being made in the work and not available from prior (extensive) literature, especially in the Introduction.

2.     The authors designed a HIV-2 Integrase model based on the SIV Intasome structure (PDB: 6RWM) and molecular docking. The authors should clarify why the available HIV-1 Intasome structure (PDB:8FN7) from (PMID: 37478179)  was not used. Besides, the binding poses of the same drug can be different between HIV and other retroviral Integrases such as PFV often used as models for the HIV Intasome significantly affecting drug resistance (PMID: 38077045). The authors can comment on this and include appropriate citations.

Minor comment:

1.     In the Introduction, the authors mention the effect of daily medication burden and compliance on HIV drug resistance. It has been shown that drug-resistance mutations can become trapped in the population leading to both transmission and persistence of resistance in HIV.

Perhaps, the authors can also provide some comment on the “evolutionary entrenchment” of drug resistance in patients and how it varies between patient populations, types, and how it affects transmission.

Reviewer 4 Report

Comments and Suggestions for Authors

Kiarie et al. used a pseudoparticles based inhibition profiling assay to compare activities of integrase inhibitors (RAL, EVG, DTG, BIC & CAB) and capsid inhibitor (LEN) at inhibiting HIV-1 and HIV-2 infections in vitro. The topic is of interest as the burden of interest as HIV-2 is largely understudied compared to HIV-1. The methodology is sound. The results presented here however are similar to previously published articles. The manuscript is well written. However, I think some information and analyses are missing, but can be further added, and the conclusions should sometimes be re-drawn, in regards to some points I mentioned below. Thus, I highlight here some major and minor points that I would like the authors to address. 

Major Comments

A.   My first point here concerns LEN and HIV-2. Data presented in Figure 1 concerning LEN are not commented in result section and the methodology behind this experiment is unclear. Indeed, the authors describe in their methods section that pseudovirions were produced either alone or in the presence of LEN, but it is unclear if data presented in Figure 1 used untreated pseudovirions or LEN treated pseudovirions. Please clarify and comment in the result section. Furthermore, why were there no condition without LEN in Figure 1 (x axis starting at 2 log PM LEN). An untreated condition is mandatory to draw further conclusions.

B.    Only two major groups of HIV-2 are currently circulating (A& B groups). ROD strain is group A, but group B could have been analyzed. More generally, only strain for HIV-1 and HIV-2 were used, limiting the extrapolation of these results. Fox example in the LEN initial study from Link et al. two HIV-2 strains and >20 strains of HIV-1 were tested. Would there be a way to increase the number of strains analyzed (for HIV-1 as well), to comfort and strengthen the results showed by the author. Moreover, the IC50 for LEN HIV-2 is 1 log lower than in the Link et al. paper, further reinforcing the need for more strains to be analyzed, and further discuss this result in the discussion section. Please comment on this discrepancy between the two studies.

C.    Concerning the pseudovirions produced in the presence of LEN, the range of LEN concentrations used is different from the one in Figure 1. and the IC50 (at least > 1 log higher for the lowest concentration). The authors should explain why these concentrations were used.

D.   It would have been of interest to analyze the docking of LEN with the HIV-2 capsid.

E.    Line 274 : Figure 3 refers to “inhibition of HIV-1 and 2 pseudovirion production by LEN”, however it does not refer to capsid production but rather to inhibition of infection with aberrant capsids produced in the presence of LEN. Please modify. Furthermore these results were not discussed in the discussion section. Please comment.

Minor Comments

1.     The authors state that the RT activity was similar with pseudoviruses produced in the presence of LEN. This data is missing and should added in a supplementary table.

2.     Line 148 “Virus” production should be replaced by ‘pseudovirions” Further occurrences in the same paragraph to be changed as well.

3.     Line 148 “HIV-2” is only mentioned in the title of the paragraph however we understand that it was done for HIV-1 as well, please add.

4.     Line 148 The “capsid formation” was not assessed, rather it was the ability of an aberrant capsid produced in presence of LEN to infect cells. Please rephrase.

5.     Why was cell viability assessed in an independent experiment and not during the inhibition profiling? 

6.     Line 176 “GFP” is mentioned but according to the other paragraphs there is no GFP encoding plasmids used in the experiments. Please clarify.

7.     Figure 2: maybe mention in the figure legend that “green” residues represent the catalytic site

8.     Some typos:

a.     Line 219: DNA (not DAN)

b.     Line 346: limited (not limed)

c.     Line 334: “f” is missing in “of”

Comments on the Quality of English Language

See above